# Real-Time Ocean Current Compensation for AUV Trajectory Tracking Control Using a Meta-Learning and Self-Adaptation Hybrid Approach

**DOI:** 10.3390/s23146417

**Published:** 2023-07-14

**Authors:** Yiqiang Zhang, Jiaxing Che, Yijun Hu, Jiankuo Cui, Junhong Cui

**Affiliations:** 1College of Computer Science and Technology, Jilin University, Changchun 130012, China; zhangyq0923@163.com (Y.Z.);; 2Shenzhen Institute for Advanced Study, University of Electronic Science and Technology of China (UESTC), Shenzhen 518110, China; 3School of Automation Science and Engineering, South China University of Technology, Guangzhou 510641, China

**Keywords:** meta-learning, adaptive control, AUV, trajectory tracking, underwater current

## Abstract

Autonomous underwater vehicles (AUVs) may deviate from their predetermined trajectory in underwater currents due to the complex effects of hydrodynamics on their maneuverability. Model-based control methods are commonly employed to address this problem, but they suffer from issues related to the time-variability of parameters and the inaccuracy of mathematical models. To improve these, a meta-learning and self-adaptation hybrid approach is proposed in this paper to enable an underwater robot to adapt to ocean currents. Instead of using a traditional complex mathematical model, a deep neural network (DNN) serving as the basis function is trained to learn a high-order hydrodynamic model offline; then, a set of linear coefficients is adjusted dynamically by an adaptive law online. By conjoining these two strategies for real-time thrust compensation, the proposed method leverages the potent representational capacity of DNN along with the rapid response of adaptive control. This combination achieves a significant enhancement in tracking performance compared to alternative controllers, as observed in simulations. These findings substantiate that the AUV can adeptly adapt to new speeds of ocean currents.

## 1. Introduction

Autonomous underwater vehicles (AUVs) have become an essential tool for humans in exploring and utilizing the underwater world. These vehicles are reliable and efficient, and they offer a wide range of applications such as seabed mapping, ocean monitoring, and submarine pipeline maintenance [1,2,3,4,5].

The inherent inaccuracies within AUV dynamic models, coupled with the temporal variations in hydrodynamic parameters due to the unpredictable and dynamic nature of ocean currents [6,7,8], pose considerable challenges in achieving accurate trajectory tracking tasks.

To enhance trajectory tracking performance in the presence of underwater currents, model-based control methods have been proposed by researchers [9,10,11,12,13]. Vu and Le Thanh et al. [10] designed a robust station-keeping (SK) control algorithm based on sliding mode control (SMC) theory to guarantee better performance of a AUV under ocean current disturbance, but they addressed the problem of stable hovering in a plane. An intelligent-PID and PD feedforward controllers hybrid controller was proposed in [13]; model-based hydrodynamics and ocean current observer were employed to contribute to the performance of the control system. However, the effect of different ocean current speeds on the parameters of the mathematical model was not considered.

Building upon this foundation, several studies have integrated adaptive approaches to address the time-varying of parameters [14,15,16]. The authors proposed a double-loop framework utilizing a backstepping strategy to decouple tracking errors in [14]. Within this framework, three adaptive robust control (ARC) strategies were developed and analyzed, each exhibiting robustness to unknown time-varying ocean currents. However, the method did not develop a suitable basis function and adaptive law, resulting in limited controller performance.

In summary, the model-based trajectory tracking control method has shown promising results, with adaptive control further addressing the problem of dynamic parameter changes. However, this approach suffers from inaccuracies in the dynamic model. For example, the mathematical model of the basis function in adaptive control is designed based on physical information related to hydrodynamics [17,18,19], and the model will no longer be accurate when physical properties such as the velocity, density of the underwater currents, and mass of AUV change.

Setting our approach apart from these methods, we suggest replacing the traditional basis function with DNN to enhance the accuracy of the dynamics model. Thanks to its powerful representational capabilities [20], it becomes possible to construct an accurate hydrodynamic model using just 10 min of AUV navigation data, improving trajectory tracking performance.

The intersection of deep learning (DL) and control theory has garnered significant interest in resolving trajectory tracking issues. Nevertheless, our focus is on holistically amalgamating DL with control theory [21,22,23,24], as opposed to an application where the controller exclusively relies on the DNN output [25,26,27,28]. Yan et al. [22] proposed an adaptive dual integral SMC for AUVs with unsure dynamics. They utilized a neural network controller fused with a conditional integrator for vehicle adaptation. However, it suffers from limited generalization and chattering reduction. Shi et al. [23] employed a DNN to model airflow disturbance during the landing phase of a drone. Despite using a method that aggregated DL and adaptive control, it inadequately addressed the issue of DNN overfitting and limited its application exclusively to aerodynamics. In contrast to these approaches, our DNN ensures generalizability by leveraging meta-learning, and its output is adjusted via an adaptive mechanism.

In light of previous discussions, we adopt self-adaption to tackle time-variant parameters and employ DNNs to overcome inaccuracies in dynamic models. Building upon this, this paper presents a meta-learning-based adaptive control (MAC) method.

The MAC comprises two distinct stages: offline meta-learning and online self-adaption. Initially, the AUV gathers data from various ocean currents, each characterized by different speeds. Using this data, a DNN is developed as the basis function and trained offline using meta-learning techniques. Through this process, the DNN successfully captures the intrinsic hydrodynamic features. Figure 1 illustrates the proposed mechanism.

It is worth noting that we want DNN to learn the common features of hydrodynamics, rather than overfitting the data sets we collected, so we introduce another discriminator network to avoid overfitting [29]. Furthermore, spectral normalization is employed in the DNN to ensure it satisfies the Lipschitz property, thereby guaranteeing the stability of the network output [30].

Subsequently, during the online phase, a set of linear coefficients is updated using the adaption law to modify the DNN output. These linear coefficients, in tandem with the DNN outputs, represent the unmodeled hydrodynamic force. Given that the DNN has learned high-order hydrodynamic features, the linear coefficients within the low-dimensional space can swiftly adapt to new ocean current speeds in real time. As a result, the predicted hydrodynamic force will be compensated for the thrusters. A flow chart of the MAC is illustrated in Figure 2.

Finally, in the Simulink and Python co-simulation, the trajectory tracking error is significantly improved.

Overall, the key contributions of this work are as follows:We employ deep learning theory to improve the adaptive control method instead of completely relying on DNN output. To the best of our knowledge, this is the first combination of meta-learning and adaption in AUV control.We evaluate the effectiveness of the proposed method by implementing it in the simulation of a fully actuated six-DOF AUV.We provide theoretical guarantees for the feasibility, stability, generalization, and low time complexity of the MAC, which ensures the practical significance of our approach.

The remaining sections of this article are organized as follows. Section 2 introduces the dynamic model of the AUV and the problem that we aim to solve in this paper. We propose the details of the meta-learning-based adaptive control method in Section 3. In Section 4, we present and discuss the simulation results from AUV trajectory tracking tasks with different control methods. Finally, Section 5 summarizes the conclusions of this research.

## 2. Modeling and Problem Formulation

### 2.1. Assumptions

To test our method, we used the underwater vehicle model developed by the MathWorks team [31]. The real-world dynamic model of the AUV is quite complex and requires many parameters. To simplify the model, the following assumptions were made:The AUV is fairly symmetrical about its three planes.The center of buoyancy of the AUV is located on the geometric symmetry plane.The AUV is considered a rigid body; thus, there are no bending or geometrical deformations.The entire AUV body is completely submerged in water.

### 2.2. Modeling

This study utilizes an AUV equipped with six thrusters to achieve six degrees of freedom control (surge, heave, sway, roll, pitch, and yaw). Each thruster is labeled from T${}_{1}$ to T${}_{6}$, as illustrated in Figure 3. The paper employs three coordinate systems, which include the body frame (b-frame), inertial reference frame (i-frame), and the current velocity frame (c-frame). It is worth noting that the c-frame, which represents the three-axis velocity of the underwater current, is omitted in Figure 3 since it coincides with the i-frame.

To consider a fully actuated AUV, first, we need to define the following three vectors [32]:(1)$$\begin{array}{cc} \eta & ={\left[\begin{array}{c} x,y,z,\phi ,\theta ,\psi \end{array}\right]}^{T},\\ \nu & ={\left[\begin{array}{c} u,v,w,p,q,r \end{array}\right]}^{T},\\ \tau & ={\left[\begin{array}{c} F_{x},F_{y},F_{z},T_{k},T_{m},T_{n} \end{array}\right]}^{T}, \end{array}$$
where *x*, *y*, and *z* are linear positions of the AUV; ϕ, θ, and ψ are the vectors of Euler angles; ν is the velocity vector; *u*, *v*, and *w* represent linear velocities; *p*, *q*, and *r* are the angular velocities; $F_{x}$, $F_{y}$, and $F_{z}$ are forces; $T_{k}$, $T_{m}$, and $T_{n}$ denote torques; and τ is a vector of forces and torques acting on the AUV. All of the above parameters are in the b-frame.

Then, we introduce the dynamics of this AUV model [31]:(2)$$\begin{array}{c} M\ddot{\eta }+C\left(\dot{\eta }\right)\dot{\eta }+D\left(\dot{\eta }\right)\dot{\eta }+G\left(\eta \right)=\tau , \end{array}$$
the kinematic equation of the AUV is given by the following formula:(3)$$\begin{array}{c} \dot{\eta }=J\left(\eta \right)\nu , \end{array}$$
where η, $\dot{\eta }$, $\ddot{\eta }$ are the position, velocity, and acceleration vectors, respectively, in the i-frame. *M* and $C(\dot{\eta })$ are the mass matrix and Coriolis effects associated with the rigid body, including added mass, and G(η) represents the combined gravitational and buoyancy effects. J(η) is the transformation matrix, which can be obtained as follows:(4)$$\begin{array}{c} J\left(\eta \right)=\left[\begin{array}{cc} J_{1}\left(\eta \right) & 0_{3\times 3}\\ 0_{3\times 3} & J_{2}\left(\eta \right) \end{array}\right], \end{array}$$
where
(5)$$\begin{array}{c} J_{1}\left(\eta \right)=\left[\begin{array}{ccc} c\psi c\theta & -s\psi c\phi +c\psi s\theta s\phi & s\psi s\phi +c\psi c\phi s\theta \\ s\psi c\theta & c\psi c\phi +s\phi s\theta c\psi & -c\psi s\phi +s\theta s\psi c\phi \\ -s\theta & c\theta s\phi & c\theta c\phi \end{array}\right], \end{array}$$
(6)$$\begin{array}{c} J_{2}\left(\eta \right)=\left[\begin{array}{ccc} 1 & s\phi t\theta & c\phi t\theta \\ 0 & c\phi & -s\phi \\ 0 & \frac{s\phi }{c\theta } & \frac{c\phi }{c\theta } \end{array}\right], \end{array}$$
where *s*(·) = sin(·), *c*(·) = cos(·), and *t*(·) = tan(·).

Moreover, the aforementioned matrices are described as follows:(7)$$\begin{array}{c} M=\left[\begin{array}{cccccc} m+X_{\dot{u}} & 0 & 0 & 0 & mz_{G} & -my_{G}\\ 0 & m+Y_{\dot{v}} & 0 & -mz_{G} & 0 & mx_{G}+Y_{\dot{r}}\\ 0 & 0 & m+Z_{\dot{w}} & my_{G} & -mx_{G}+Z_{\dot{q}} & 0\\ 0 & -mz_{G} & my_{G} & I_{xx}+K_{\dot{p}} & I_{xy} & I_{xz}\\ mz_{G} & 0 & -mx_{G}+M_{\dot{w}} & I_{yx} & I_{yy}+M_{\dot{q}} & I_{yz}\\ -my_{G} & mx_{G}+N_{\dot{v}} & 0 & I_{zx} & I_{zy} & I_{zz}+N_{\dot{r}} \end{array}\right], \end{array}$$
(8)$$\begin{array}{c} C\left(\dot{\eta }\right)=\left[\begin{array}{c} C_{1},C_{2},C_{3},C_{4},C_{5},C_{6} \end{array}\right], \end{array}$$
where
(9)$$\begin{array}{c} C_{1}=[0,0,0,-m\left(y_{G}q+z_{G}r\right),m\left(x_{G}q-w\right)-Z_{\dot{w}}w,m\left(x_{G}r+v\right)+Y_{\dot{v}}{v]}^{T},\\ C_{2}=[0,0,0,m\left(y_{G}q+w\right)+Z_{\dot{u}}w,-m\left(z_{G}r+x_{G}p\right),m\left(y_{G}r-u\right)-X_{\dot{u}}{u]}^{T},\\ C_{3}=[0,0,0,m\left(z_{G}p-v\right)-Y_{\dot{v}}v,m\left(z_{G}q+u\right)+X_{\dot{u}}u,-m\left(x_{G}p+y_{G}q\right){]}^{T},\\ C_{4}=[m\left(y_{G}q+z_{G}r\right),-m\left(y_{G}q+w\right)-Z_{\dot{w}}w,-m\left(z_{G}P-v\right)+Y_{\dot{v}}v,\\ \;\;\;\;\;\;\;\;\;\;0,-I_{yz}q+I_{xz}p-I_{zz}r-N_{\dot{r}}r,-I_{yz}r-I_{xy}p+I_{yy}q+M_{\dot{q}}{q]}^{T},\\ C_{5}=[-m\left(x_{G}q-w\right)+Z_{\dot{w}}w,m\left(z_{G}r+x_{G}p\right),-m\left(z_{G}q+u\right)-X_{\dot{u}}u,\\ \;\;\;\;\;\;\;\;\;\;-I_{yz}q-I_{xz}p+I_{zz}r+N_{\dot{r}}r,0,I_{xz}r+I_{xy}q-I_{xx}p-K_{\dot{p}}{p]}^{T},\\ C_{6}=[-m\left(x_{G}r+v\right)-Y_{\dot{v}}v,-m\left(y_{G}r-u\right)+X_{\dot{u}}u,m\left(x_{G}p+y_{G}q\right),\\ \;\;\;\;\;\;\;\;\;\;I_{yz}r+I_{xy}p-I_{yy}q-M_{\dot{q}}q,-I_{xz}r-I_{xy}q+I_{xx}p+K_{\dot{p}}{p,0]}^{T}, \end{array}$$
(10)$$\begin{array}{c} G\left(\eta \right)=\left[\begin{array}{c} (W-B)\sin\theta \\ -(W-B)\cos\theta \sin\phi \\ -(W-B)\cos\theta \cos\phi \\ -\left(y_{G}W-y_{b}B\right)\cos\theta \cos\phi +\left(z_{G}W-z_{b}B\right)\cos\theta \sin\phi \\ \left(z_{G}W-z_{b}B\right)\sin\theta +\left(x_{G}W-x_{b}B\right)\cos\theta \cos\phi \\ -\left(x_{G}W-x_{b}B\right)\cos\theta \sin\phi -\left(y_{G}W-y_{b}B\right)\sin\theta \end{array}\right]. \end{array}$$

All symbols of variables used in the above equations can be explained as follows: *m* denotes the mass of the AUV; *W* and *B* are the weight of the AUV body and the submerged buoyancy force expressed in the i-frame; $x_{G}$, $y_{G}$, and $z_{G}$ are the center of gravity; $x_{b}$, $y_{b}$, and $z_{b}$ are the center of buoyancy; $I_{xx}$, $I_{yy}$, $I_{zz}$, $I_{xy}=I_{yx}$, $I_{xz}=I_{zx}$, and $I_{yz}=I_{zy}$ are the moments of inertia of the AUV; $X_{\dot{u}}$, $Y_{\dot{v}}$ v, $Z_{\dot{w}}$, $K_{\dot{p}}$, $M_{\dot{q}}$, and $N_{\dot{r}}$ are partial derivative coefficients [33].

$D(\dot{\eta })$ is the damping matrix, which includes the drag $F_{d}$ and lift forces $F_{l}$ [34]. $F_{d}$ and $F_{l}$ represent the outcomes of pressure and friction induced by the fluid flowing around the surface of the vehicle. These forces hinge significantly on various complex factors: the shape of vehicle, the skin friction, and specifically the velocity and density of the underwater currents. The calculation for $F_{d}$ and $F_{l}$ is as follows [35]:(11)$$\begin{array}{c} F_{d}=\frac{1}{2}\rho V^{2}C_{d}A, \end{array}$$
(12)$$\begin{array}{c} F_{l}=\frac{1}{2}\rho V^{2}C_{l}A, \end{array}$$
where ρ is the density of the fluid, *V* is the vehicle velocity, and *A* is a cross-section reference area. Drag and lift coefficients $C_{d}$ and $C_{l}$ are normally obtained from laboratory experiments [36]. More details about the matrices and parameters above in the simulation model can be found in [31].

In this work, we define $D\left(\dot{\eta }\right)\dot{\eta }$ as the hydrodynamic effect that will be learned by the DNN, which is equivalent to general force $f_{a}={\left[f_{a,x},f_{a,y},f_{a,z}\right]}^{T}$; then, a new AUV dynamic model is given by
(13)$$\begin{array}{c} M\ddot{\eta }+C\left(\dot{\eta }\right)\dot{\eta }+G\left(\eta \right)+f_{a}\left(\eta ,\dot{\eta },c\right)=\tau , \end{array}$$
where *c* is an unknown hidden state used to represent the currents at different speeds, and the control law is based on Equation (Equation 13).

### 2.3. DNN Input and Output

Hydrodynamic force $f_{a}$ is represented by non-modeling methods in this work and is decomposed as
(14)$$\begin{array}{c} f_{a}\left(\eta ,\dot{\eta },c\right)\approx \bar{\phi }\left(\eta ,\dot{\eta }\right)a\left(c\right), \end{array}$$
where $\bar{\phi }$ is the basis function, and $a\in R^{3}$ are linear coefficients associated with the *c*. The decomposition in Equation (Equation 14) is feasible, and $\bar{\phi }$ can be approximated by DNN [37,38].

To consider an *L*-layer DNN $\phi \left(x;\Theta_{1}\right):R^{9}\to R^{3\times 3}$ parameterized by weights $\Theta_{1}=\left\{W^{1},\cdots ,W^{L+1}\right\}$, which is used to replace the basis function $\bar{\phi }$, we have
(15)$$\begin{array}{c} f_{a}\left(\eta ,\dot{\eta },c\right)\approx \phi \left(x;\Theta_{1}\right)a\left(c\right), \end{array}$$
with $x=\left[v,E,f_{u}\right]$, where *v*, *E*, and $f_{u}$ represent the linear velocity, Euler angles, and resultant forces of the thrusters in the i-frame, respectively. The output of ϕ is a matrix representing prior information of the hydrodynamic model.

### 2.4. Meta-Learning Goal

The goal of offline meta-learning is to solve the optimization problem
(16)$$\begin{array}{c} \min_{\Theta_{1},a}\sum_{k=1}^{K}\sum_{i=1}^{N}{∥J_{k}^{\left(i\right)}∥}^{2}, \end{array}$$
with $J_{k}^{\left(i\right)}=y_{k}^{\left(i\right)}-\phi \left(x_{k}^{\left(i\right)};\Theta_{1}\right)a_{k}$, where *y*, the label for the neural network, represents the observed value of $f_{a}$ as computed from Equation (Equation 13), *i* is the ordinal number of the sample, k(K=5) is the current condition index, and $∥\cdot ∥$ denotes binary norm.

### 2.5. Adaptive Control Goal

In the online adaptive stage, we aim to constantly adjust *a* to stabilize the system to a desired trajectory and minimize the root mean squared error (RMSE) defined by
(17)$$\begin{array}{c} \mathrm{RMSE}=\frac{1}{T}\sum_{t=1}^{T}∥\eta^{\left(t\right)}-\eta_{d}^{\left(t\right)}∥, \end{array}$$
where *t* is the time step.

## 3. Meta-Learning-Based Adaptive Control

### 3.1. Assumptions

In this work, simulation scenarios with different current speeds need to be built. To efficiently verify the effectiveness of the MAC, we make the following simplifying assumptions:The direction of currents comes from the positive *y*-axis in the c-frame, but a small range of disturbance is added to the three axes in the c-frame.Disturbance is consistent with additive white Gaussian noise (AWGN) with a variance of 0.01.

The details of ocean currents simulation can be found in Section 4.

### 3.2. Data Collection

Position, Euler angles, velocities, and accelerations of the AUV are gathered using a baseline controller on a random trajectory at 0, 0.8, 1.5, 2, and 2.3 m/s current speeds for 10 min each, as shown in Figure 1. Each randomly selected trajectory is designed by us to ensure that the AUV can adequately collect data in the ocean currents. The collected data set is $D=\left\{D_{1},\ldots ,D_{K}\right\}$, where
(18)$$\begin{array}{c} D_{k}=\left\{x_{k}^{\left(i\right)},y_{k}^{\left(i\right)}\right\}, \end{array}$$

Figure 4 shows that the data distribution changes dramatically between data subsets, which are very suitable as training data for meta-learning so that the DNN can adapt to a new current speed.

### 3.3. Preparation for Meta-Learning

The Lipschitz constant serves to constrain the gradient of the function. Certain studies have demonstrated that DNN stability can be assured by employing spectral normalization to uphold Lipschitz continuity within the DNN [30]. The activation function for $\phi (x;\Theta_{1})$ in this study is ReLU, which abides by Lipschitz continuity. Thus, our focus is solely on constraining the parameter matrix of each layer. Spectral normalization is employed in the weight matrices in every layer during the training process as indicated below:(19)$$\begin{array}{c} \bar{W}=W/\sigma \left(W\right)\cdot \gamma^{\frac{1}{L+1}}, \end{array}$$
where σ(·) stands for the function determining the maximum singular value of the matrix, and γ represents the anticipated Lipschitz constant for the DNN. Having secured stable output from the DNN, we proceed to address the overfitting concern to ensure the DNN accurately learns the authentic hydrodynamic physical characteristics.

To prevent data overfitting under a specific current speed, an improved optimization problem, grounded in Equation (Equation 16), is proposed:(20)$$\begin{array}{c} \max_{\Theta_{2}}\min_{\Theta_{1},a}\sum_{k=1}^{K}\sum_{i=1}^{N}\left({∥J_{k}^{\left(i\right)}∥}^{2}-\alpha \cdot \mathrm{loss}\left(h(\phi \left(x_{k}^{\left(i\right)};\Theta_{1}\right);\Theta_{2}),k\right)\right), \end{array}$$
where *h* is another DNN parameterized by $\Theta_{2}$, which works as a discriminator to predict the index out of *K* current conditions, loss(·) is cross-entropy function, and α is a hyperparameter to control the degree of regularization. The updated objective function can suppress ϕ from overfitting the data set.

### 3.4. Meta-Learning

Fitting a dataset with a consistent data distribution using a DNN represents a relatively straightforward task. However, as illustrated in Figure 4, substantial variations exist within the distributions of each dataset. As a result, a DNN trained on one dataset may demonstrate unsatisfactory performance when applied to other datasets, given its limited generalizability [39]. Even when uniting all datasets and conducting joint DNN training, challenges may still arise during testing, particularly when encountering unfamiliar current speeds, as the datasets may not comprehensively represent all possible speeds.

We tackle this problem by using meta-learning and other DL techniques, which allow the DNN to learn a real hydrodynamic model instead of overfitting a certain dataset.

In this work, meta-learning does not refer to the specific DNN structure, but to a training procedure, as shown in Algorithm 1.
**Algorithm** **1** Meta-Learning Algorithm1: **procedure** Meta-Learning($\left\{D_{1},\ldots ,D_{K}\right\}$)2:       **Initialize:** DNN weights $\Theta_{1}$, $\Theta_{2}$3:       **repeat**4:             **for** each $D_{k}$ in *D* **do**5:                   Randomly sample $D_{a}$, $D_{tr}$6:                   Solve the least squares problem using $D_{a}$7:                   $a\leftarrow {\left(\Phi^{⊤}\Phi \right)}^{-1}\Phi^{⊤}Y$8:                   Fix *h* and train ϕ using $D_{tr}$ with loss9:                   $\sum_{i=1}^{N}\left({∥J_{k}^{\left(i\right)}∥}^{2}-\alpha \cdot \mathrm{loss}\left(h(\phi \left(x_{k}^{\left(i\right)}\right)),k\right)\right)$10:                 Fix ϕ and train *h* using $D_{tr}$ with loss11:                 $\sum_{i=1}^{N}\mathrm{loss}\left(h(\phi \left(x_{k}^{\left(i\right)}\right)),k\right)$12:            **end for**13:      **until** converged14:**end procedure**

Instead of training the entire dataset, we take turns training subsets of data with different current speeds, then randomly sample $\left\{D_{a},D_{tr}\right\}$ from subset $D_{k}$, where $D_{a}$ is the adaption set, $D_{tr}$ is the training set, and $D_{a}\cap D_{tr}=\emptyset$.

Instead of fitting the current forces directly, before training we solve for the coefficients *a* using the least squares method, as follows:(21)$$\begin{array}{c} a={\left(\Phi^{⊤}\Phi \right)}^{-1}\Phi^{⊤}Y, \end{array}$$
where $\Phi ={\left[\begin{array}{c} \phi \left(x_{1}\right),\phi \left(x_{2}\right),\cdots ,\phi \left(x_{M}\right) \end{array}\right]}^{⊤}$, $Y={\left[\begin{array}{c} y_{1},y_{2},\cdots ,y_{M} \end{array}\right]}^{⊤}$, and *M* is the size of $D_{a}$. Then, ϕ and *h* are trained alternatively using data from $D_{tr}$, and stochastic gradient descent (SGD) updates the parameters in each iteration.

Noting that *a* as an adaptive parameter is a function of ϕ in the training process, the gradient concerning the parameters in DNN ϕ will backpropagate through *a*.

### 3.5. Adaptive Control

In this part, the adaptive controller is designed for Equation (Equation 13) based on DNN. We define the composite variable as follows:(22)$$\begin{array}{c} s=\dot{\tilde{\eta }}+\Lambda \tilde{\eta }=\dot{\eta }-v_{d}, \end{array}$$
where position tracking error $\tilde{\eta }=\eta -\eta_{d}$; $\eta_{d}$ and $\nu_{d}$ denote the desired position and reference velocity vectors, respectively; and Λ is a positive definite or diagonal matrix. Now the trajectory tracking problem is transformed to tracking a reference velocity $v_{d}={\dot{\eta }}_{d}-\Lambda \tilde{\eta }$.

The control law and adaption are designed as follows:(23)$$\begin{array}{c} \tau =M{\dot{v}}_{d}+C\left(\dot{\eta }\right)v_{d}+G\left(\eta \right)+\phi \left(\eta ,\dot{\eta }\right)\hat{a}-Ks, \end{array}$$
(24)$$\begin{array}{c} \dot{\hat{a}}=-P\left(\phi^{⊤}s+\phi^{⊤}e_{h}+\beta \hat{a}\right), \end{array}$$
where $\hat{a}$ represents an estimated parameter, *P* is another positive definite matrix, *K* and β are gain, and $e_{h}$ is the hydrodynamic prediction error of the DNN, defined as follows:(25)$$\begin{array}{c} e_{h}=\phi \hat{a}-f_{a}. \end{array}$$

To define representation error $e_{r}=\phi a-f_{a}$, and parameter error $\tilde{a}=\hat{a}-a$, the prediction error can be written as
(26)$$\begin{array}{c} e_{h}=e_{r}+\phi \tilde{a}. \end{array}$$

Taking Equation (Equation 26) into Equation (Equation 24), and combining Equation (Equation 23) with the dynamics in Equation (Equation 13), the closed loop dynamics can be obtained as follows:(27)$$\begin{array}{c} M\dot{s}=-\left(C+K\right)s+\phi \tilde{a}+e_{r}, \end{array}$$
(28)$$\begin{array}{c} P^{-1}\dot{\tilde{a}}=-\phi^{⊤}s-\left(\phi^{⊤}\phi +\beta I\right)\tilde{a}-\left(\phi^{⊤}e_{r}+\beta a+P^{-1}\dot{a}\right), \end{array}$$

The matrix form can be derived as follows:(29)$$\begin{array}{cc} \left[\begin{array}{cc} M & 0\\ 0 & P^{-1} \end{array}\right]\left[\begin{array}{c} \dot{s}\\ \dot{\tilde{a}} \end{array}\right]= & \left[\begin{array}{cc} -(C+K) & \phi \\ -\phi^{⊤} & -(\phi^{⊤}\phi +\beta I) \end{array}\right]\left[\begin{array}{c} s\\ \tilde{a} \end{array}\right]\\ & +\left[\begin{array}{c} e_{r}\\ -(\phi^{⊤}e_{r}+\beta a+P^{-1}\dot{a}) \end{array}\right]. \end{array}$$

**Theorem** **1.**
*If the dynamics of the vehicle can be described by Equation (Equation 13), which uses the control law in Equation (Equation 23) and adaption law in Equation (Equation 24), considering the closed-loop system in Equation (Equation 29), the AUV can follow the desired trajectory with bounded error with a suitable choice of the design constants K, P, and β.*


**Proof.** Considering the Lyapunov function *V* given by
(30)$$\begin{array}{cc} V & ={\left[\begin{array}{c} s\\ \tilde{a} \end{array}\right]}^{⊤}\left[\begin{array}{cc} M & 0\\ 0 & P^{-1} \end{array}\right]\left[\begin{array}{c} s\\ \tilde{a} \end{array}\right], \end{array}$$
to use the closed loop dynamics given in Equation (Equation 29) and the skew-symmetric property of $\dot{M}-2C$, we obtain the derivative of *V* as follows:
(31)$$\begin{array}{cc} \dot{V}= & {\left[\begin{array}{c} s\\ \tilde{a} \end{array}\right]}^{⊤}\left[\begin{array}{cc} M & 0\\ 0 & P^{-1} \end{array}\right]\left[\begin{array}{c} \dot{s}\\ \dot{\tilde{a}} \end{array}\right]+{\left[\begin{array}{c} s\\ \tilde{a} \end{array}\right]}^{⊤}\left[\begin{array}{cc} \dot{M} & 0\\ 0 & \frac{d}{dt}\left(P^{-1}\right) \end{array}\right]\left[\begin{array}{c} s\\ \tilde{a} \end{array}\right]\\ = & 2{\left[\begin{array}{c} s\\ \tilde{a} \end{array}\right]}^{⊤}\left[\begin{array}{cc} -(C+K) & \phi \\ -\phi^{⊤} & -(\phi^{⊤}\phi +\beta I) \end{array}\right]\left[\begin{array}{c} s\\ \tilde{a} \end{array}\right]+2{\left[\begin{array}{c} s\\ \tilde{a} \end{array}\right]}^{⊤}\left[\begin{array}{c} e_{r}\\ -(\phi^{⊤}e_{r}+\beta a+P^{-1}\dot{a}) \end{array}\right]\\ & +{\left[\begin{array}{c} s\\ \tilde{a} \end{array}\right]}^{⊤}\left[\begin{array}{cc} \dot{M} & 0\\ 0 & \frac{d}{dt}\left(P^{-1}\right) \end{array}\right]\left[\begin{array}{c} s\\ \tilde{a} \end{array}\right]\\ = & 2{\left[\begin{array}{c} s\\ \tilde{a} \end{array}\right]}^{⊤}\left[\begin{array}{cc} -K & \phi \\ -\phi^{⊤} & -(\phi^{⊤}\phi +\beta I) \end{array}\right]\left[\begin{array}{c} s\\ \tilde{a} \end{array}\right]+2{\left[\begin{array}{c} s\\ \tilde{a} \end{array}\right]}^{⊤}\left[\begin{array}{c} e_{r}\\ -(\phi^{⊤}e_{r}+\beta a+P^{-1}\dot{a}) \end{array}\right]\\ & +{\left[\begin{array}{c} s\\ \tilde{a} \end{array}\right]}^{⊤}\left[\begin{array}{cc} 0 & 0\\ 0 & \frac{d}{dt}\left(P^{-1}\right) \end{array}\right]\left[\begin{array}{c} s\\ \tilde{a} \end{array}\right]\\ = & -2{\left[\begin{array}{c} s\\ \tilde{a} \end{array}\right]}^{⊤}\left[\begin{array}{cc} K & 0\\ 0 & \frac{d}{dt}\left(P^{-1}\right)-\left(\phi^{⊤}\phi +\beta I\right) \end{array}\right]\left[\begin{array}{c} s\\ \tilde{a} \end{array}\right]\\ & +2{\left[\begin{array}{c} s\\ \tilde{a} \end{array}\right]}^{⊤}\left[\begin{array}{c} e_{r}\\ -(\phi^{⊤}e_{r}+\beta a+P^{-1}\dot{a}) \end{array}\right]\\ \leq & -2\lambda_{1}V+2\lambda_{2}\sqrt{V}, \end{array}$$
where
(32)$$\begin{array}{c} \lambda_{1}=\frac{\lambda_{\min}\left(\tilde{H}\right)}{\lambda_{\max}\left(H\right)}, \end{array}$$
(33)$$\begin{array}{c} \lambda_{2}=\sqrt{1/\lambda_{\min}\left(H\right)}∥\left[\begin{array}{c} e_{r}\\ \phi^{⊤}e_{r}+\beta a+P^{-1}\dot{a} \end{array}\right]∥ \end{array}$$
by defining
(34)$$\begin{array}{c} H=\left[\begin{array}{cc} M & 0\\ 0 & P^{-1} \end{array}\right] \end{array}$$
and
(35)$$\begin{array}{c} \tilde{H}=\left[\begin{array}{cc} K & 0\\ 0 & \frac{d}{dt}\left(P^{-1}\right)-\left(\phi^{⊤}\phi +\beta I\right) \end{array}\right]. \end{array}$$Let $U=\sqrt{V}=\sqrt{y^{⊤}Hy}$ with $y=\left[\begin{array}{c} s\\ \tilde{a} \end{array}\right]$. It can be derived that $\dot{V}=2U\dot{U}$. Then, a differential inequality is obtained from Equation (Equation 31), as
(36)$$\begin{array}{c} \dot{U}+\lambda_{1}U\leq \lambda_{2}, \end{array}$$
such that
(37)$$\begin{array}{c} ∥y∥\leq ∥y\left(0\right)∥e^{-\lambda_{1}t}+\frac{\lambda_{2}}{\lambda_{1}}\left(1-e^{-\lambda_{1}t}\right). \end{array}$$Finally, *y* converges to the bounded error exponentially. This completes the proof. ☐

## 4. Simulation Results and Discussion

In this section, we establish a simulation to assess the performance of the MAC method. We have realized the meta-learning algorithm in Python 3.7, and employed the Simulink component in MATLAB R2021b to execute the AUV simulation. During the online simulation stage, the controller in Simulink can call upon the DNN model, trained in Python, in real time. The entire simulation is conducted on a laptop computer, equipped with a Windows 11 operating system and a x64 processor.

### 4.1. DNN Training

The hyperparameters of the DNN are shown in Table 1, where *k* and $\bar{k}$ are the size of $D_{a}$ and $D_{tr}$, respectively; *l* and $\bar{l}$ are the learning rate of *h* net and ϕ net, respectively; α is the coefficient of the regular term; γ is the maximum norm of *a*; *q* represents the update frequency of *h* net; $S_{N}$ is the maximum single-layer spectral norm of ϕ; and $N_{E}$ is the number of training epochs. In Table 1, the value of the structure of the neural network ϕ indicates that ϕ has six layers of neurons (including input and output layers), and the number of neurons in each layer is indicated. The same applies for neural network *h*.

During the experiment, we notice that most parameters are insensitive, but *k* affected the convergence of network ϕ. When k=64, the prediction error could converge quickly to a stable range, as shown in Figure 5. After the completion of network training, we obtained good prediction results. The prediction of the hydrodynamic force at 1.5 m/s ocean current is shown in Figure 6, which proves that the underlying physical features of hydrodynamics are successfully captured by ϕ net.

Moreover, to substantiate the practicability of the technique proposed in this study within a real system, we undertook an analysis of the time consumed during DNN prediction. The computational complexity of neural networks has often been subject to criticism. Contrarily, this study introduces an approach wherein an offline model is utilized, substantially reducing the need for online training time. Additionally, our DNN is characterized by a mere four hidden layers and no more than 50 neurons in each layer, thus significantly minimizing computational power requirements. As illustrated in Figure 7, the maximum time taken for each prediction does not exceed 0.03 s, averaging around 0.016 s. In actual control systems, a response frequency of 20 Hz is adequate to sustain normal operations, thus affirming the practical relevance of our DNN methodology.

### 4.2. Simulate the Effects of Ocean Currents on AUV

The real ocean current model is very complex, and therefore it is difficult to model the exact real ocean current in a simulation environment. In this paper, the ocean current velocity is set on the three axes of the c-frame to simulate the ocean current effect, and we assume that the disturbance satisfies the AWGN to ensure that the current speed is not fixed.

As can be seen in Figure 8, the flow direction of the current is in the positive direction of the y-axis in c-frame with following settings: $V_{x}$ = 0 m/s, $V_{y}$ = 1.2 m/s, and $V_{z}$ = 0 m/s.

To achieve the forward motion of the AUV, we set the thrust forces on two main thrusters as T${}_{1}$ = 10 N and T${}_{2}$ = 10 N, respectively, while all thrusters, T${}_{3}$, T${}_{4}$, T${}_{5}$, and T${}_{6}$, are set as 0 N. The simulation duration is 120 s, and the sampling time is 0.05 s. Under these ocean current configurations, as can be observed in Figure 9, the AUV without current effect moves forward in a straight line but deviates slightly due to the torque caused by the thrusters in the water, and the effect of ocean currents can cause the AUV to deviate significantly from the forward direction.

Furthermore, from Figure 10 and Figure 11, it can be seen that the position, orientation, and velocities of the AUV can be seriously affected by the ocean currents during forward motion. For example, the roll and pitch motions of the AUV appeared to be significantly oscillatory; a perturbation can be observed within the first 20 s in Figure 11.

### 4.3. Control Performance

The simulation has demonstrated the impacts of ocean currents on the motion of the AUV. For this section, we employ an ocean current setting of 1.2 m/s and design an S-shaped trajectory tracking task to validate the effectiveness of the MAC in simulation. It is important to underscore that this trajectory and the selected current speed are novel to the AUV and are not encompassed within the training dataset.

To establish a reliable benchmark, we conducted tests on dozens of sets of PID parameters and selected the five most representative groups: PID1, PID2, PID3, PID4, and PID5. The details of the PID implementation can be found in [40]. The AUV initial point is $\left(x_{0},y_{0},z_{0}\right)=\left(0,-20,5\right)$; the starting point and the ending point of the trajectory are (0,0,−5) and (388,−20,45), respectively. Figure 12 displays the trajectories of the AUV in a 3D plane for five different groups of PID parameters. Convergence of tracking error can be clearly observed in Figure 13.

Using Equation (Equation 17), RMSE can be calculated as shown in Table 2. Table 2 shows that PID1 outperforms the other PID controllers. Therefore, PID1 is chosen as the baseline controller.

In addition, we designed variants of the MAC for comparison, including methods called adaptive and meta-learning. Both methods utilize similar control laws to MAC from Equation (Equation 23), but with some differences. Specifically, in the adaptive method, ϕ represents conventional hydrodynamics, as presented in [32]. Meanwhile, in the meta-learning method, ϕ follows the same approach as MAC, but instead of calculating the linear coefficient $\hat{a}$, it is a constant determined by the mean value of $\hat{a}$ calculated by the MAC method. In other words, the meta-learning method mainly depends on the output of the neural network. MAC is a combination of both, using the neural network ϕ output, while being able to adaptively adjust the coefficient $\hat{a}$.

In the same underwater current and task settings, the S-shaped tracking trajectories of all methods are shown in Figure 14. It can be observed from Figure 14 that all methods except for meta-learning are able to track the predetermined trajectory effectively. The trajectory tracking details of each method are presented in Figure 15. Obviously, the error of the meta-learning method fails to converge within 600 s, resulting in an RMSE of 19.317, which is significantly higher than that of the other methods. The adaptive method is able to track the trajectory with an RMSE of 4.987 and maintain the error within a smaller range. However, its performance is still inferior to that of the baseline controller. The baseline rapidly approaches the trajectory and stabilizes the error at approximately 1 m, resulting in an RMSE of 1.998 over the entire process. Furthermore, due to the accurate prediction of the hydrodynamic force and fast adaption, the error of the MAC method stabilizes around 0.5 m. The MAC method exhibits an RMSE of 1.373 m over a duration of 600 s, outperforming the baseline method, which relies solely on error feedback, and the adaptive method, grounded on inaccurate hydrodynamics. This results in an enhancement in tracking performance by approximately 31.3% and 72.5%, respectively.

In addition to inspecting the trajectory tracking error, we delve deeper to compare each method in greater detail. Based on Equation (Equation 23), the thrust compensation for the MAC is obtained by multiplying the DNN output with the adaptive coefficients. The generalized forces of the thrusters can be observed in Figure 16, where the bow horizontal thruster T${}_{5}$ and the stern horizontal thruster T${}_{6}$ inputs are compensated with MAC for counteracting excessive ocean currents. In contrast, the meta-learning method exhibits a limited ability to efficiently adapt the DNN output based on the error signal. Moreover, its input of thruster T${}_{5}$ and T${}_{6}$ remains relatively stable, which poses challenges to effectively track intricately curved trajectories. Figure 17 illustrates that the MAC facilitates faster changes in the yaw angle ψ. During AUV turns, Figure 18 demonstrates that the MAC provides faster linear velocity in the y-direction, enhancing the AUV performance in cornering.

The variation of the vector $\hat{a}$ for the three methods is shown in Figure 19, where a${}_{1}$, a${}_{2}$, and a${}_{3}$ represent the coefficients in the x, y, and z directions in i-frame, respectively. This demonstrates how the adaptive and MAC methods dynamically adjust the vector $\hat{a}$ to compensate for the sea current forces and achieve good results in trajectory tracking. However, Table 3 reveals that the tracking performance of MAC is superior, indicating that the neural network model is more accurate and precise than the traditional modeling approach, as it is fitted with a large amount of data.

On the other hand, as the meta-learning method uses a fixed value of $\hat{a}$, it fails to compensate for the thrust force accurately. The above discussion proves that the combination of meta-learning and self-adaption is superior to either method used independently.

This section presents a simulation study of the AUV trajectory tracking task in the presence of ocean current disturbance, and we demonstrate the dynamic behaviors of the AUV and the changing process of adaptive parameters and tracking errors. In a comparison of the adaptive, meta-learning, and baseline controllers, the results show that MAC effectively uses DNN and self-adaption to track trajectories more stably and accurately. Moreover, MAC successfully adapts to new ocean conditions, as evidenced by its ability to track a current speed of 1.2 m/s.

## 5. Conclusions

To address the challenges of model inaccuracy and time-varying parameters posed by the dynamic ocean current, we proposed a hybrid method comprising offline meta-learning and online self-adaptation. We utilized a DNN to represent the hydrodynamic basis function and adjust the coefficients linearly to fit the unmodeled hydrodynamics in real time. This combination facilitated compensation, enabling the controller to respond quickly and accurately to new current speeds while maintaining stability, interpretability, and generalization.

We have implemented and tested this controller in our simulation. Based on the results, it is apparent that the MAC significantly enhances the tracking error, stabilizing it at approximately 0.5 m in a new ocean current environment, which is much better than the compared methods. As a result, it demonstrates a remarkable improvement in tracking accuracy when compared to both the adaptive approach and learning-based approach. This improvement confirms that the AUV has successfully adapted to the new current speed.

In future work, we will further investigate the deep-learning-based trajectory tracking method for AUVs in ocean currents, with particular emphasis on addressing adaptability to currents of varying directions and velocities. Simultaneously, the structure of the neural network needs to be traded off to ensure that the controller has sufficient response frequency. To further expand our research, we plan to validate our control method by conducting tests on an actual AUV.

## Figures and Tables

**Figure 1 sensors-23-06417-f001:**
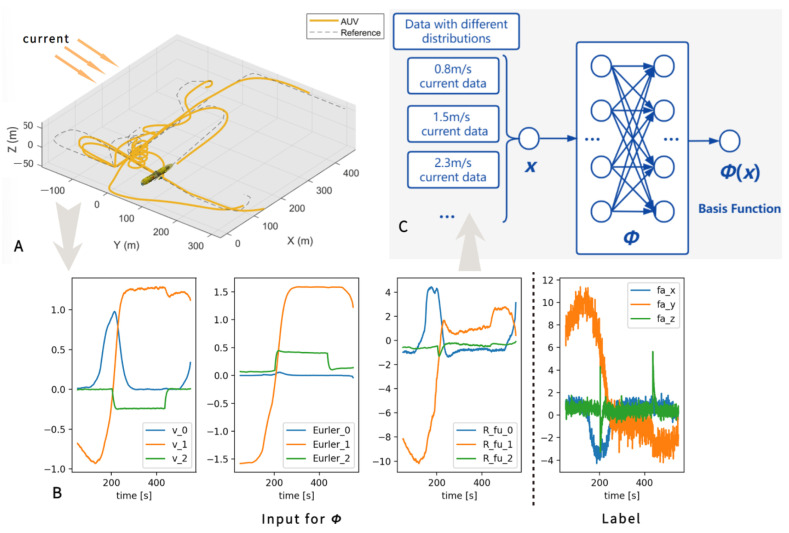
Data collection and training. (**A**) AUV travels at different speeds to collect data. (**B**) The data serve as inputs and labels to DNN. (**C**) DNN learns the underlying physical features of hydrodynamic force.

**Figure 2 sensors-23-06417-f002:**
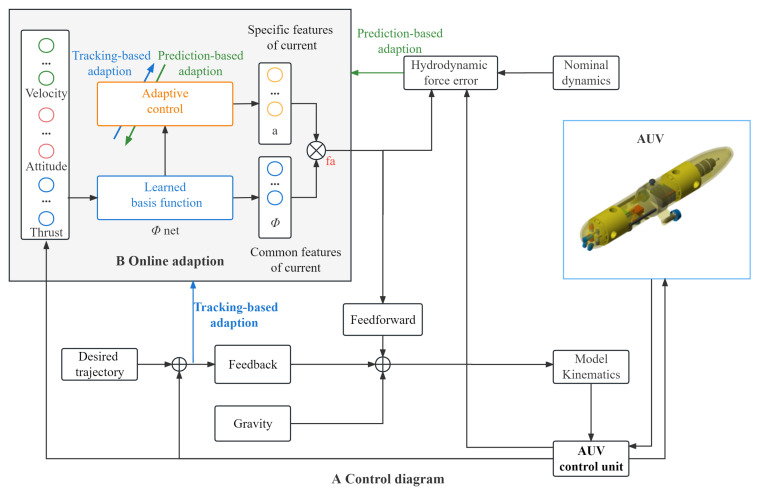
Meta-learning-based adaptive controller design. (**A**) Overview diagram of control method. (**B**) The online adaptation block in the adaptive controller.

**Figure 3 sensors-23-06417-f003:**
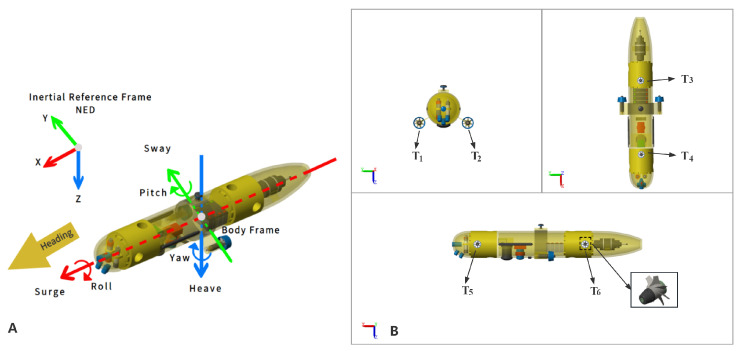
Simulation model of AUV rigid body. (**A**) Schematic presentation of an AUV with a different frame of reference. (**B**) Three views of AUV.

**Figure 4 sensors-23-06417-f004:**
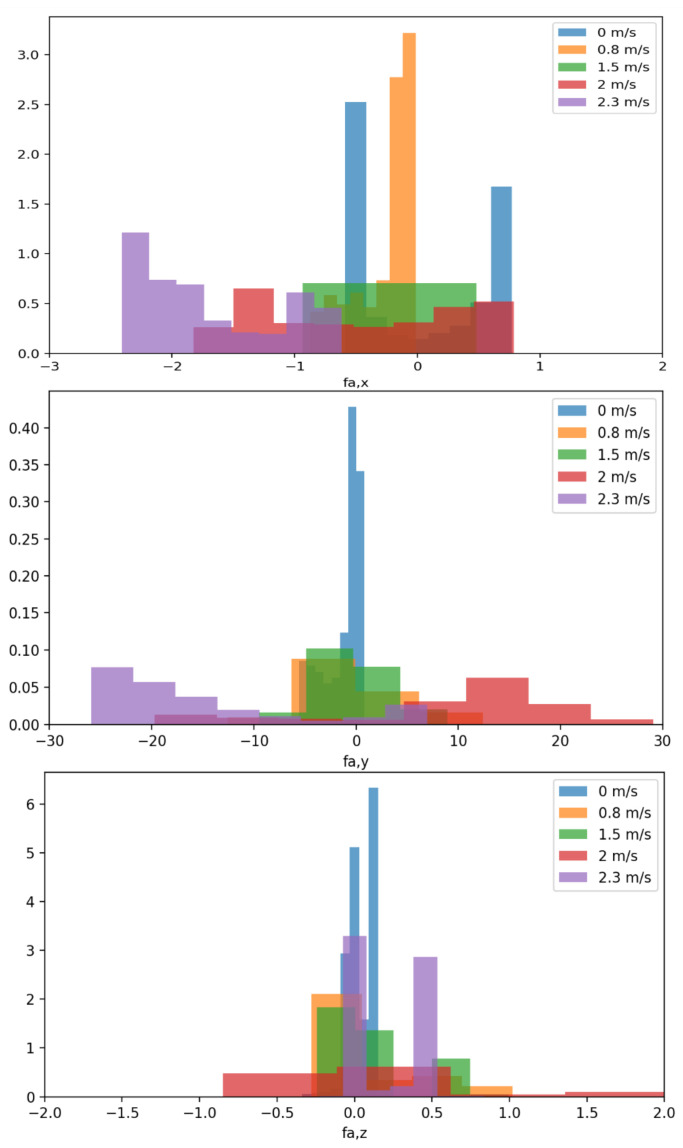
Data distribution of different current speeds. The three figures represent the hydrodynamic forces in the three dimensional axes: *x*, *y*, and *z*, respectively. Additionally, in each direction, we computed the probability density distribution of the hydrodynamic force corresponding to different velocities.

**Figure 5 sensors-23-06417-f005:**
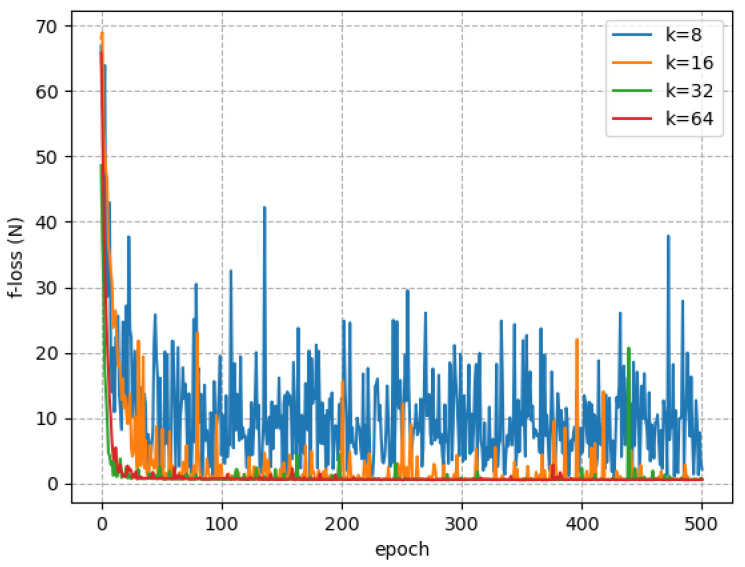
Error convergence for different *k*.

**Figure 6 sensors-23-06417-f006:**
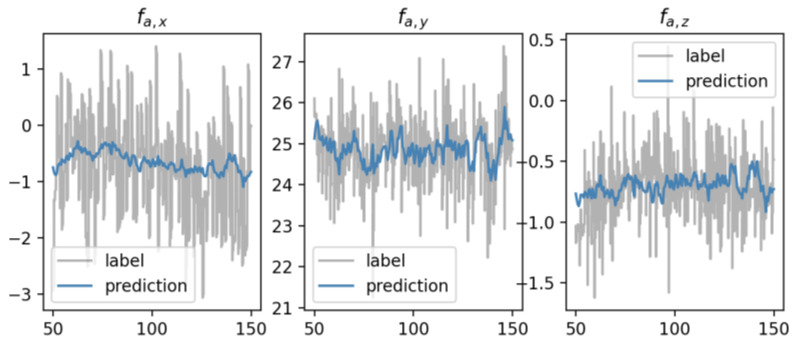
Predictions of hydrodynamic force $f_{a}$ at 1.5 m/s ocean current. With offline training, the hydrodynamic force of the three axes can be successfully modeled.

**Figure 7 sensors-23-06417-f007:**
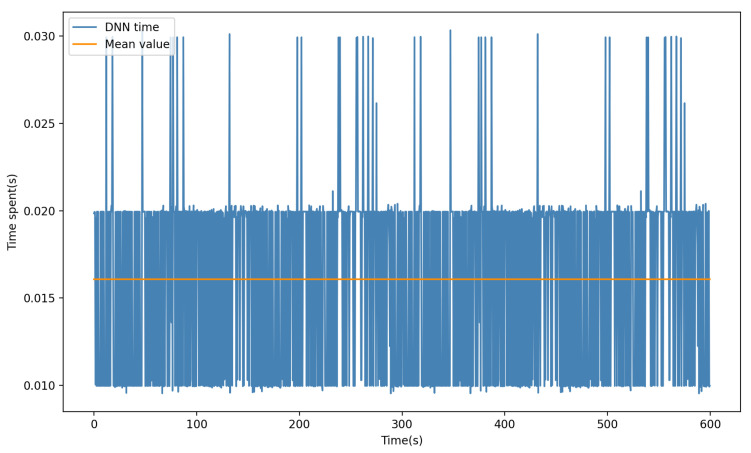
DNN time spent per prediction. The figure illustrates the time taken by the neural network to make each prediction within a span of 600 s, generally ranging between 0.01 to 0.03 s.

**Figure 8 sensors-23-06417-f008:**
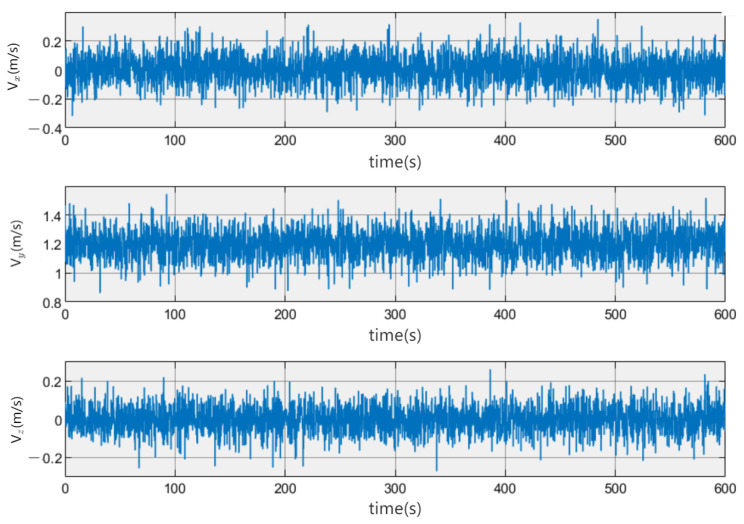
The three-axis velocity of the underwater current. The figure depicts that the speed of the current in each direction of the coordinate axis experiences minor fluctuations.

**Figure 9 sensors-23-06417-f009:**
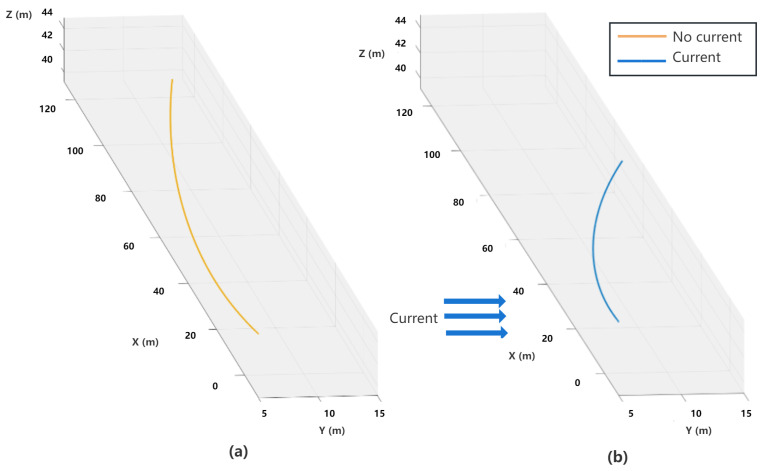
Trajectories of the AUV in forward motion: (**a**) without current effects and (**b**) with current effects.

**Figure 10 sensors-23-06417-f010:**
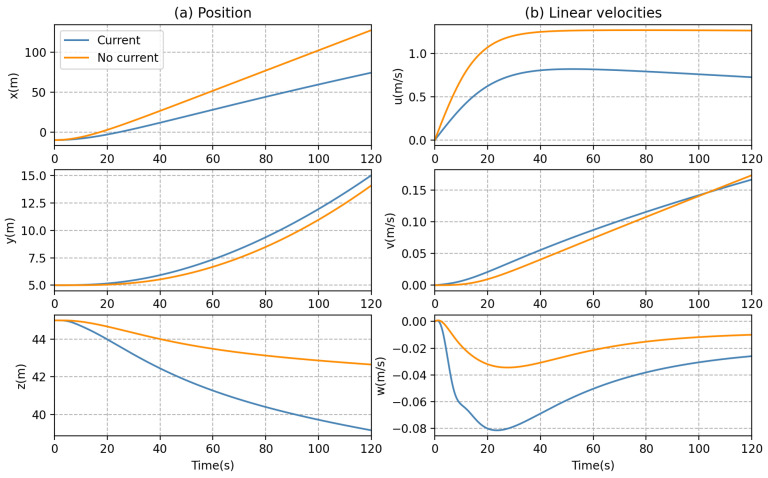
AUV dynamic behaviors in forward motion: (**a**) position and (**b**) linear velocities.

**Figure 11 sensors-23-06417-f011:**
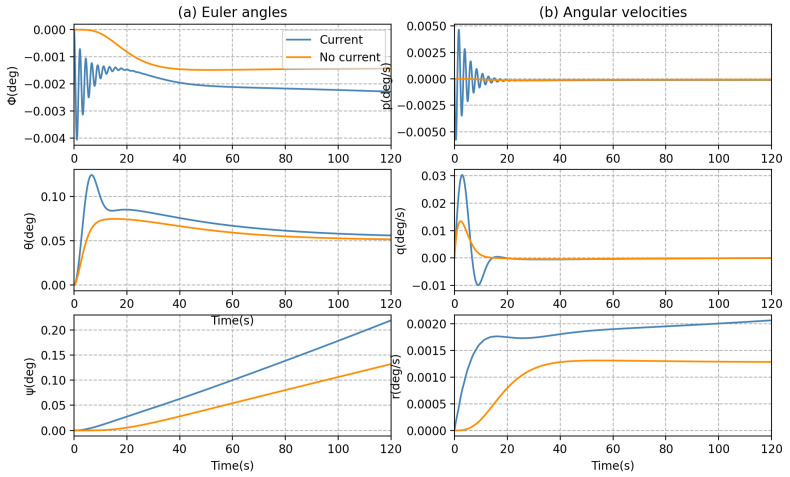
AUV dynamic behaviors in forward motion: (**a**) Euler angles and (**b**) angular velocities.

**Figure 12 sensors-23-06417-f012:**
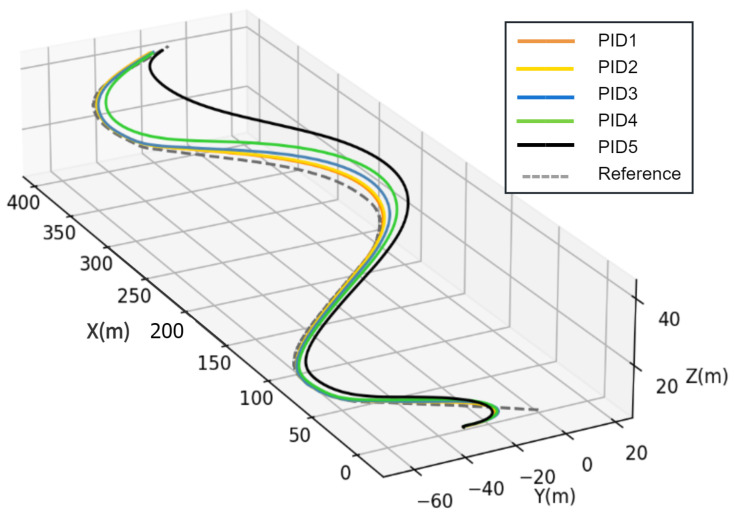
Trajectories of the AUV with five groups of PID parameters.

**Figure 13 sensors-23-06417-f013:**
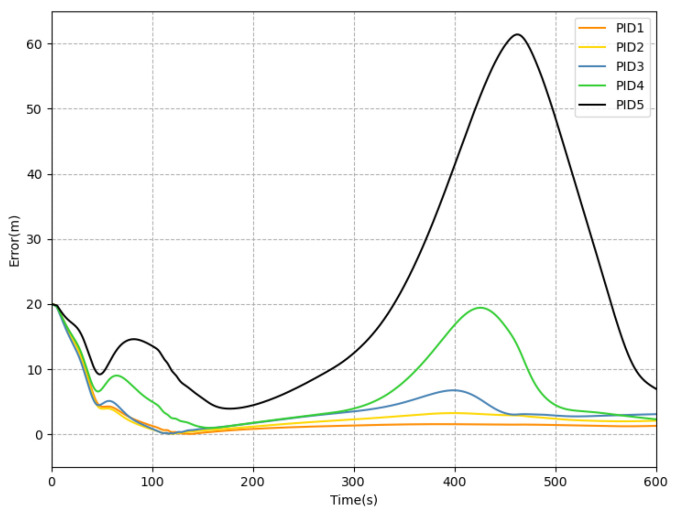
Tracking error of reference trajectory with PID.

**Figure 14 sensors-23-06417-f014:**
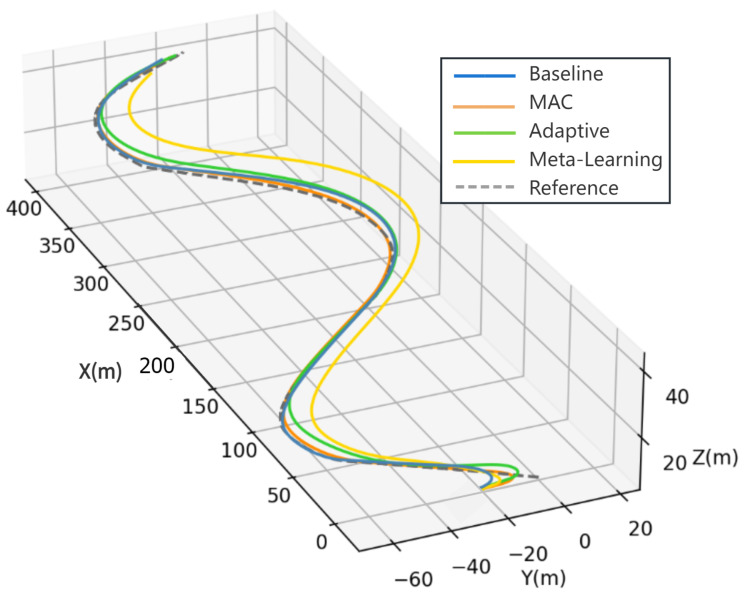
The trajectory for each method.

**Figure 15 sensors-23-06417-f015:**
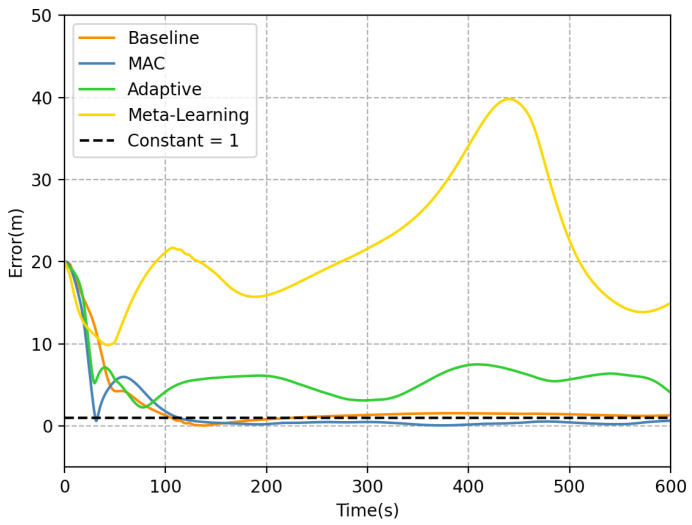
The tracking error for each method.

**Figure 16 sensors-23-06417-f016:**
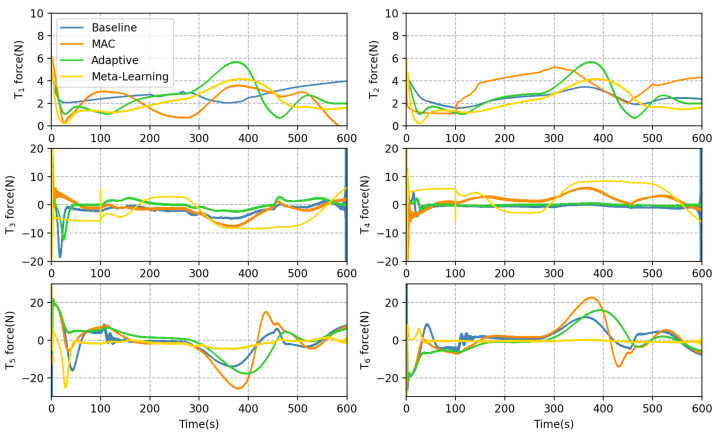
Input for each thruster.

**Figure 17 sensors-23-06417-f017:**
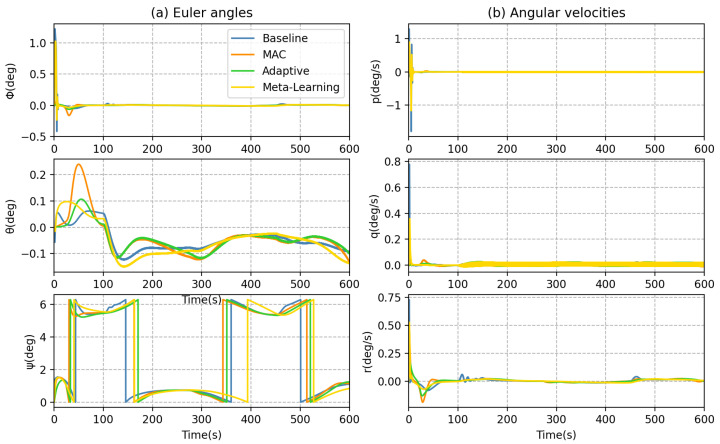
AUV dynamic behaviors: (**a**) Euler angles and (**b**) angular velocities.

**Figure 18 sensors-23-06417-f018:**
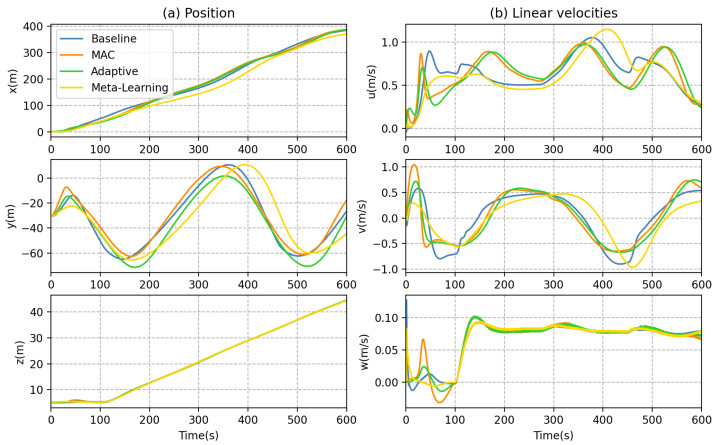
AUV dynamic behaviors: (**a**) position and (**b**) linear velocities.

**Figure 19 sensors-23-06417-f019:**
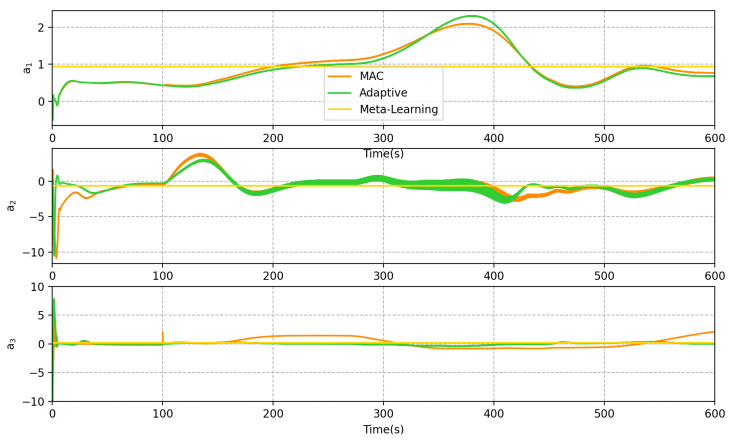
Variation of linear coefficient $\hat{a}$. $\hat{a}$ represents the vector, which is multiplied with ϕ to express the hydrodynamic force. The figure illustrates the variation across each of its components.

**Table 1 sensors-23-06417-t001:** Hyperparameters in DNN.

Hyperparameter	Value
Architecture of ϕ net	9→30→40→50→30→3
Architecture of *h* net	3→20→5
*k*	64
$\bar{k}$	256
α	0.1
*l*	0.001
$\bar{l}$	0.0005
γ	15
*q*	0.5
$S_{N}$	2
$N_{E}$	500

**Table 2 sensors-23-06417-t002:** Statistics of tracking errors in different directions of each PID.

Methos	$E_{x}$	$E_{y}$	$E_{z}$	RMSE	Unit
PID1	1.157	1.629	0.016	1.998	m
PID2	1.679	1.961	0.017	2.582	m
PID3	1.893	2.984	0.021	3.533	m
PID4	2.381	6.161	0.184	6.605	m
PID5	12.175	15.844	0.045	19.982	m
Average	3.857	5.716	0.057	6.94	m

**Table 3 sensors-23-06417-t003:** MAC and baseline tracking error statistics in different directions.

Methos	$E_{x}$	$E_{y}$	$E_{z}$	RMSE	Unit
Baseline	1.157	1.629	0.016	1.998	m
MAC	0.596	1.235	0.058	1.373	m
Adaptive	3.599	3.452	0.048	4.987	m
Meta-Learning	15.732	11.209	0.062	19.317	m

## Data Availability

The data presented in this paper are available after contacting the corresponding author.

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
