# Peer review of "Real-Time Ocean Current Compensation for AUV Trajectory Tracking Control Using a Meta-Learning and Self-Adaptation Hybrid Approach"

_sensors, 2023, doi:10.3390/s23146417_

Round 1
Reviewer 1 Report
The paper does not provide significant experimental details needed to correctly assess its contribution: What is the validation procedure used?
Quality of figures is so important too. Please provide some high-resolution figures. Some figures have a poor resolution.
Conclusion should state scope for future work.
Authors should add the parameters of the proposed deep learning algorithms.
No recent references exist to support and contrast the work in terms of AUV Trajectory Tracking Control
Explain why the current method was selected for the study, its importance and compare with traditional methods.
What is the motivation of the proposed work? Research gaps, objectives of the proposed work should be clearly justified.
Check the mathematical notation of the whole paper.
What assumptions authors made during the simulation phase of this research work? Provide a critique on this aspect.
The research results reported are too premature for publication. More work is needed to substantiate the conclusions in your manuscript.
The paper also contains a lot of typos that must be revised.
The obtained results must be compared with the results that would be obtained with existing techniques to justify the superiority
The language usage throughout this paper need to be improved, the author should do some proofreading on it.
Reviewer 2 Report
The paper proposes a novel method to control AUVs. To my understanding, the process includes running the AUV in different speeds and observe the damping matrix in the dynamic model, training a DNN with the data to get fa, and finally running the adaptive controller with fa in the dynamic model. Here are my comments:
1、 Why do you just learn damping matrix and not others?
2、 To collect data, it require the AUV to run in different speeds and collect the position, Euler angles, velocities, and accelerations to observe the damping matrix. It is an easy job in simulation. But have you considered how to do this in practical? How to handle the measure errors of the sensors?
3、 A baseline controller is applied when collecting data. Will this controller affect he final result?
4、 The value of architecture in table 1 seem to be the number of neurons. It should have some description in the context.
5、 I still can’t quite understand how the second DNN works. I think one DNN may be enough. Please make it more clear.
6、 There should be some data about the speed of the DNN。
7、 Please make it clear that how the currents are added to the simulation model. Do they directly added to the speed vector of the AUV?
8、 I think Fig. 11 and Fig. 12 are not necessary since we all know that the performance can be improved by adjusting PID values.
9、 The format of the reference should be carefully checked. For example, Reference 25 do not label the resource.
The spelling and grammar should be carefully checked. The following is part of the problems.
propose utilizing in line 38
models inaccuracy and parameters variation in line 56
differing speeds, then to in line 62
is linear coefficients in line 145
y as label for neural network is the observed in line 154
while testing when facing new current in line 196
hydrodynamics modeling in line 295
Reviewer 3 Report
The authors of the entitled manuscript "Real-Time Ocean Current Compensation for AUV Trajectory Tracking Control using a Meta-Learning and Self-Adaptation Hybrid Approach" presented an adaptive control approach to resolve the issues of varying hydrodynamic parameters using deep learning approach to improve its inaccuracy.
1. Abstract is well presented. However, the introduction section requires more literature to support the study. I found the majority of the article from Ocean Engineering.
2. Line 136: Cd and Cl can be obtained computationally. In other words this study could be approached using CFD modelling.
3. All figures need attention such as the title for each figure. Also. Try to zoom in Figure 16. Also, add (a) (b) and (c) to each figure description. Expand the description of the figures.
4. Expand the explanation of Figure 4. (Lines 174)
5. Line: 242. Remove the box after "This completes the proof."
6. How long took the simulation? This point is very important as the specification of the laptop should be added here. This is one of the limitations of this study as mentioned in the conclusion section.
7. Figures 9 and 10: why the W(m/s) and r (rad/s) look like inverse.
8. Please elaborate on the novelty of the work, so that it is clear which gap you are addressing. This is a just mathematical model.
9. The English language needs proof-reading by a professional service.
Overall, this is a good topic. In saying that, I would like the authors to address the above points.
The English language needs proof-reading by a professional service.
Round 2
Reviewer 1 Report
The obtained results must be compared with the results that would be obtained with existing techniques to justify the superiority
Make the conclusion more systematic and add future scope to this paper.
Explain novelty of your work presented in this work.
The paper also contains a lot of typos that must be revised.
Reviewer 3 Report
I would like to acknowledge the authors. Good paper but future work is needed to test the outcome physically.
